# Docosahexaenoic Acid and Arachidonic Acid Levels Are Associated with Early Systemic Inflammation in Extremely Preterm Infants

**DOI:** 10.3390/nu12071996

**Published:** 2020-07-05

**Authors:** Ann Hellström, William Hellström, Gunnel Hellgren, Lois E. H. Smith, Henri Puttonen, Ing-Marie Fyhr, Karin Sävman, Anders K. Nilsson, Susanna Klevebro

**Affiliations:** 1Institute of Neuroscience and Physiology, Sahlgrenska Academy at University of Gothenburg, 40530 Gothenburg, Sweden; ann.hellstrom@medfak.gu.se (A.H.); gunnel.hellgren@gu.se (G.H.); anders.k.nilsson@gu.se (A.K.N.); 2Department of Paediatrics, Institute of Clinical Sciences, Sahlgrenska Academy, University of Gothenburg, 41686 Gothenburg, Sweden; william.hellstrom@gu.se (W.H.); henri.puttonen@vgregion.se (H.P.); karin.savman@pediat.gu.se (K.S.); 3Institute of Bioscience, Sahlgrenska Academy at University of Gothenburg, 40530 Gothenburg, Sweden; 4Department of Ophthalmology, Boston Children’s Hospital, Harvard Medical School, Boston, MA 02115, USA; Lois.Smith@childrens.harvard.edu; 5Department of Pathology, Region Västra Götaland, Sahlgrenska University Hospital, 41345 Gothenburg, Sweden; ifyhr@me.com; 6Department of Neonatology, Region Västra Götaland, the Queen Silvia Children’s Hospital, Sahlgrenska University Hospital, 41345 Gothenburg, Sweden; 7Department of Clinical Science and Education, Stockholm South General Hospital, Karolinska Institutet, 11883 Solna, Sweden

**Keywords:** inflammation, preterm infant, polyunsaturated fatty acid, interleukin-6

## Abstract

Fetal and early postnatal inflammation have been associated with increased morbidity in extremely preterm infants. This study aimed to demonstrate if postpartum levels of docosahexaenoic acid (DHA) and arachidonic acid (AA) were associated with early inflammation. In a cohort of 90 extremely preterm infants, DHA and AA in cord blood, on the first postnatal day and on postnatal day 7 were examined in relation to early systemic inflammation, defined as elevated C-reactive protein (CRP) and/or interleukin-6 (IL-6) within 72 h from birth, with or without positive blood culture. Median serum level of DHA was 0.5 mol% (95% CI (confidence interval) 0.2–0.9, *P* = 0.006) lower than the first postnatal day in infants with early systemic inflammation, compared to infants without signs of inflammation, whereas levels of AA were not statistically different between infants with and without signs of inflammation. In cord blood, lower serum levels of both DHA (correlation coefficient −0.40; *P* = 0.010) and AA (correlation coefficient −0.54; *p* < 0.001) correlated with higher levels of IL-6. Levels of DHA or AA did not differ between infants with and without histological signs of chorioamnionitis or fetal inflammation. In conclusion, serum levels of DHA at birth were associated with the inflammatory response during the early postnatal period in extremely preterm infants.

## 1. Introduction

Fetal inflammation, as well as early postnatal inflammation, have been associated with increased mortality, and both short- and long term morbidity in preterm infants [1,2,3,4]. Polyunsaturated fatty acids have several important structural and functional roles in the body and bioactive fatty acid metabolites are of importance in immune/inflammation system regulation [5,6]. After preterm birth, placental transfer of LCPUFAs (long-chain polyunsaturated fatty acid) is abruptly interrupted. Preterm infants are able to synthesize DHA and AA from their C18 counterparts (α-linolenic and linoleic acid, respectively), albeit the endogenous synthesis rate is believed to be too low to meet infants’ requirements [7,8], and preterm infants are at risk of dysregulated inflammation.

Supplementation with omega-3 LCPUFAs during pregnancy reduces the risk of preterm birth [9], potentially by a reduced inflammatory response [10]. Studies of postnatal supplementation with omega-3 have shown heterogeneous results [11,12,13,14,15]. We performed a randomized trial comparing morbidity in extremely preterm infants receiving a parenteral olive oil-based lipid solution (without omega-3 LCPUFA) to a solution containing fish oil with omega-3 LCPUFAs. There were no differences in morbidity between the treatment groups [16]. In a secondary analysis, lower postnatal levels of AA were associated with a higher risk of developing retinopathy of prematurity (ROP) [17].

The present study aimed to investigate if serum levels of DHA and AA, in cord blood and during the first postnatal day, were associated with inflammatory conditions during the perinatal and early postnatal periods in extremely preterm infants.

## 2. Materials and Methods

### 2.1. Setting and Study Design

This is an exploratory analysis of infants included in the randomized controlled trial Donna Mega, NCT02760472 registered at ClinicalTrials.gov, described in detail in Najm et al. [16]. Briefly, inclusion criterion was gestational age (GA) less than 28 weeks, and exclusion criterion was major congenital malformations. The trial was conducted at Sahlgrenska University Hospital in Gothenburg, Sweden, between April 2013 and September 2015. Infants were randomized to either a parenteral lipid solution containing 15% fish oil with omega-3 LCPUFAs (SMOFlipid, Fresenius Kabi) or an olive oil-based lipid solution (Clinoleic, Baxter) without omega-3 LCPUFAs. Primary outcome of the study was ROP. Infants were randomized before parenteral (intravenous) lipid treatment was started. Lipid solution was initially given at 1 g/kg/day as soon as a central line was secured, normally within 6–12 H after birth, and advanced at a rate of 1 g/kg/day to a final parenteral intake of 2–3 g/kg/day. Minimal enteral feeding was initiated within 3 h of birth and feeds were administered every 2–3 h (1–2 mL/meal) with a gradual increase in volume. Maternal or donor breast milk was used throughout the study.

### 2.2. Sample Collection and Definitions

Venous blood samples were obtained when the central line was inserted before the start of parenteral lipid use on postnatal day one, and on postnatal days 7, 14, and 28 after the onset of parenteral lipid use. After centrifugation, samples were stored at −80 °C until assayed. Cord blood, as well as tissue samples from the placenta and cord, were collected at birth when possible depending on logistics. Tissue samples were obtained from umbilical cord (proximal and distal samples), roll of chorioamniotic membranes, umbilical cord insertion, and full-thickness samples of placenta. Histological examination of the placenta was performed following the College of American Pathologists guidelines and findings were classified according to the ELGAN protocol [18]. Histological chorioamnionitis (HCA) was defined as a maternal inflammatory response with neutrophil infiltration of subchorionic space, chorionic plate or amnion while fetal inflammatory response syndrome (FIRS) was defined as inflammation of the umbilical cord (funisitis) and/or neutrophilic infiltration of fetal stem vessels. The diagnoses of HCA and FIRS were based on joint analyses by two trained perinatal pathologists.

Early systemic inflammation was defined as CRP > 20 mg/L or IL-6 > 1000 pg/mL in any clinical sample within the first 72 h from birth, with or without positive blood culture. The cut-offs were based on clinical guidelines for suspected infection in the immediate postnatal period. During this study, IL-6 and CRP were routinely obtained in the clinic during insertion of the central line and additional sampling commonly occurred within the first 72 h from birth. Of the 90 infants included in this study, 85 had measurements of CRP and IL-6 the first postnatal day and all infants had at least one measurement of CRP or IL-6 within the first 72 h from birth, Appendix A. Postnatal levels of IL-6 and CRP, as well as clinical data, were collected from hospital records. GA at birth was based on ultrasound dating or on the date of last menstrual period if no ultrasound had been performed. Small for gestational age (SGA) was defined as birth weight below two standard deviation scores [19]. Sepsis was defined as clinical symptoms of infection in combination with blood culture positive for bacteria or fungus. If the blood culture indicated coagulase-negative staphylococci or a mixed bacterial flora, either an elevated CRP (>20 mg/ L) or an elevated IL-6 (>1000 pg/ mL) was required. ROP was classified according to the international classification [20].

### 2.3. Fatty Acid Analyses and Measurement of IL-6 in Cord Blood

Serum lipids were extracted [21] and fractionated on a single aminopropyl cartridge (Sep-Pak, Waters Corporation, Milford, MA, USA) to obtain a phospholipid fraction. After derivatization with hydrochloric acid in methanol, fatty acid methyl esters were measured by gas chromatography–mass spectrometry [16]. Fatty acid levels are expressed as molar percentages.

Serum concentrations of IL-6 in cord blood were measured in duplicate by Bio-plex Pro in a human cytokine 17-plex assay (Bio-Rad Laboratories, Hercules, CA, USA) according to the manufacturer’s protocol. Before analyses, all samples were diluted 1:4. Beads and antibodies within the assay were from the same batches, and all assays were run by the same operator. The lowest level of quantification was 0.09 pg/mL. Samples under the quantification limit were set lowest calibration point divided by 4 (0.09 pg/mL). The inter-assay coefficient of variation was 24.4% at 2.5 pg/mL, 10.8% at 955 pg/mL, and 11.5% at 4294 pg/mL, respectively.

### 2.4. Statistical Methods

All analyses were performed using Stata versions IC 14 or SE 16 (StataCorp LP, College Station, TX, USA). Comparisons of numerical measures were performed using quantile regression with bootstrapped confidence interval [22]. Spearman’s correlation was used to test for correlation between continuous variables. To evaluate the association between levels of fatty acid and postnatal systemic inflammation, univariable and multivariable analyses were performed using logistic regression. Box-Tidwell Transformation Test was used to test for linearity in the logit. Gestational age, SGA, preeclampsia, and mode of delivery were considered potential confounders and were included in the multivariable model. In all analyses, *p* values of <0.05 were considered significant. The study was conducted in accordance with the Declaration of Helsinki and parents/guardians gave their informed consent for inclusion. The protocol was approved by the Ethics Committee in Gothenburg (Dnr 303-11).

## 3. Results

### 3.1. Characteristics of the Study Population

In total 90 infants were randomized and included in this trial, Appendix A. Characteristics of the study infants are presented in Table 1. Samples from cord blood were collected from 40 infants. Infants for whom cord blood samples were missing were more immature and had lower birth weights compared to infants where cord blood had been obtained. Median GA was 25.0 weeks compared to 25.9 weeks (*P* = 0.004) and median birth weight BW was 693 g compared to 840 g (*P* = 0.010). Data regarding the histological diagnosis of chorioamnionitis and fetal inflammation were available from 78 infants. Serum levels of some of the major PUFAs in cord blood, on postnatal day one, 7 and day 8–28 are shown in Appendix A.

### 3.2. Docosahexaenoic Acid, Arachidonic Acid, and Early Postnatal Systemic Inflammation

In total, 23 infants (26%) had early systemic inflammation, and 4/23 had positive blood cultures. Serum levels of DHA in cord blood and during the first postnatal day were lower in infants with early systemic inflammation, but at postnatal day seven, no differences remained. Differences in levels of AA between infants with and without early systemic inflammation were not statistically significant (Table 2). Every 0.1 unit (mol%) of increase in serum level of DHA the first postnatal day was associated with reduced odds of early systemic inflammation, OR 0.91 (95% CI 0.85 to 0.97; *P* = 0.006). Adjustment for gestational age, SGA, preeclampsia and mode of delivery did not alter this finding, OR 0.90 (95% CI 0.83 to 0.97; *P* = 0.007).

Of infants with early systemic inflammation, 5/23 died before term age compared to 7/67 of infants without early systemic inflammation. Of surviving infants, 12/18 compared to 18/60 had at least one episode of sepsis (clinical symptoms and positive blood culture) before term age, and 10/18 compared to 21/60 developed severe ROP (stage 3 or more).

### 3.3. Docosahexaenoic Acid and Arachidonic Acid in Cord Blood and Signs of Fetal Inflammation

In cord blood, lower serum levels of both DHA and AA were associated with higher levels of IL-6 (Figure 1). Serum levels of DHA and AA in cord blood indicated a similar pattern as in infants with early systemic inflammation with lower levels in infants who had HCA or FIRS, but the differences were small and not statistically significant (Table 3).

## 4. Discussion

In this study, we demonstrated that levels of the omega-3 LCPUFA DHA the first postnatal day were lower in extremely preterm infants with early systemic inflammation compared to infants without systemic inflammation. We also demonstrated that low levels of both DHA and the omega-6 LCPUFA AA were associated with high levels of the pro-inflammatory cytokine IL-6 in cord blood.

LC-PUFAs of the omega-6 and omega-3 series might influence immune system regulation through several mechanisms, such as alterations in cell signaling pathways, cell membrane composition, gene expression, metabolite production, and mediation of oxidative stress [5,6,23]. The fatty acids in the omega-6 series mainly have functions in the pro-inflammatory response, whereas the fatty acids in the omega-3 series have functions important in inflammation resolution [23]. The omega-6 fatty acid AA is the precursor for both pro-inflammatory and pro-resolving metabolites [24]. IL-6 is produced in both infectious and noninfectious inflammatory conditions. It is highly expressed, easy to detect in the peripheral circulation, and widely used as an inflammatory biomarker [25]. Experimental models have demonstrated that administration of DHA reduce the expression of IL-6 [26,27].

Fetal levels of fatty acids are determined by maternal supply, placental metabolism, and transfer [28]. In a large study of DHA supplementation during pregnancy, the authors concluded that factors that explain the variation of DHA in cord blood are mainly unknown [29]. Chorioamnionitis and inflammation might influence placental fatty acid metabolism. Plasma levels of IL-6 in pregnant women had a negative correlation with placental lipoprotein activity, but in cell culture of human trophoblast cells, IL-6 stimulated fatty acid accumulation [30]. Fatty acid metabolism might also be disturbed due to a catabolic state. In intrauterine growth restricted fetuses, the maternal fetal ratio of DHA and AA was lower compared to inappropriate for gestational age fetuses [31].

Although inflammation might influence fetal levels of fatty acids, most studies argue that omega-3 fatty acids have the potential to attenuate the neonatal inflammatory response. Haghiac et al. showed that supplementation with the omega-3 LCPUFAs DHA and eicosapentaenoic acid (EPA) during pregnancy is associated with reduced expression of pro-inflammatory cytokines in the placenta [32]. Another study showed lower maternal levels of IL-6 in women who consumed a diet enriched in EPA, but no differences in cord blood cytokines [33]. Gold et al. demonstrated, in a cohort of mostly term infants, that higher levels of EPA and AA in cord plasma were associated with reduced proliferation and cytokine expression in antigen-stimulated lymphocytes. A comparison between Japanese preterm infants fed a soy-based lipid solution and Australian preterm infants fed a lipid solution containing omega-3 LCPUFAs, showed differences between the cohorts in the levels of some of the relevant non-esterified fatty acids and also in their downstream oxidized metabolites [34].

The association between higher infant levels of DHA the first postnatal day and reduced risk of early systemic inflammation has, to our knowledge, not been previously demonstrated among preterm infants. Fares et al. showed that preterm infants with late-onset sepsis, defined as clinical symptoms and positive blood culture after 72 h, had lower DHA the first postnatal day [35]. Higher linoleic acid to DHA ratio, during the period after the first postnatal week, was also associated with an increased risk of late-onset sepsis in another study [36]. Studies of supplementation with omega-3 fatty acids during pregnancy have not shown differences in the rate of sepsis among the offspring [9]. Most studies of prenatal supplementation have not evaluated preterm infants separately. Previous studies of parenteral lipid solutions containing omega-3 fatty acids have not demonstrated differences in the rate of sepsis [13]. Most studies of pre- and postnatal supplementation of LCPUFAs have not included systemic inflammation as an outcome.

A heightened inflammatory response is thought to be an important part of the pathogenesis of several neonatal morbidities. Both fetal inflammation and early neonatal inflammatory conditions have been associated with increased risk of short term morbidities as well as long term impairments [2,4,37,38,39,40].

One limitation of this study was the low number of cord blood samples. Histological diagnoses of chorioamnionitis or fetal inflammation were not significantly associated with levels of DHA or AA in our study. This could be due to differences in inflammation origin or mechanisms compared with early postnatal inflammation or an effect of small sample size. There was a discrepancy between infants with histological diagnoses of chorioamnionitis or fetal inflammation and early systemic inflammation that could indicate a difference between the conditions. The association between IL-6 and LCPUFAs in cord blood might reflect a different type of inflammation affecting infants in the first postnatal days or other mechanisms in relation to the transition to extrauterine life. In a study of lipid mediators in amniotic fluid, samples from patients with clinical chorioamnionitis had higher levels of IL-6 and lower levels of inflammation resolving mediators derived from omega-3 fatty acids compared to patients without clinical chorioamnionitis [41]. Quantification of non-esterified fatty acid could have provided additional information to our analyses, but it is reasonable to believe that serum phospholipid levels can be used as a surrogate reflecting tissue levels of DHA and AA in preterm infants. Another limitation of this study is the lack of information regarding the maternal levels of fatty acids and cytokines during pregnancy. We do not know if maternal levels of DHA were lower during pregnancy in infants who developed early systemic inflammation, or if the levels were reduced in association with birth.

## 5. Conclusions

As we have previously reported, infants receiving SMOFlipid showed increased serum levels of DHA but lower AA compared to infants receiving Clinoleic [16], and high levels of AA were associated with less ROP [17]. The results in this study also support the hypothesis that increasing both DHA and AA might be beneficial to the infant. Fetal levels of DHA and AA are affected by maternal dietary intake, enzymatic activity regulating fatty acid metabolism, and placental transport [42]. After birth, infant levels are affected by the composition of omega-6 and omega-3 fatty acids in parenteral and enteral supply [43,44]. Recent recommendations state that AA should be provided to preterm infants along DHA [45]. As the exposure during both the fetal and neonatal periods are significant for infant development, it is urgent to determine the optimal composition of fatty acid exposures during both these time periods.

This study adds to the evidence that DHA and AA are integrated parts of the inflammatory response during fetal and early postnatal life. It also highlights the importance of taking the entire prenatal to neonatal period into consideration while studying the complex interactions of fatty acids, inflammation, and neonatal outcomes.

## Figures and Tables

**Figure 1 nutrients-12-01996-f001:**
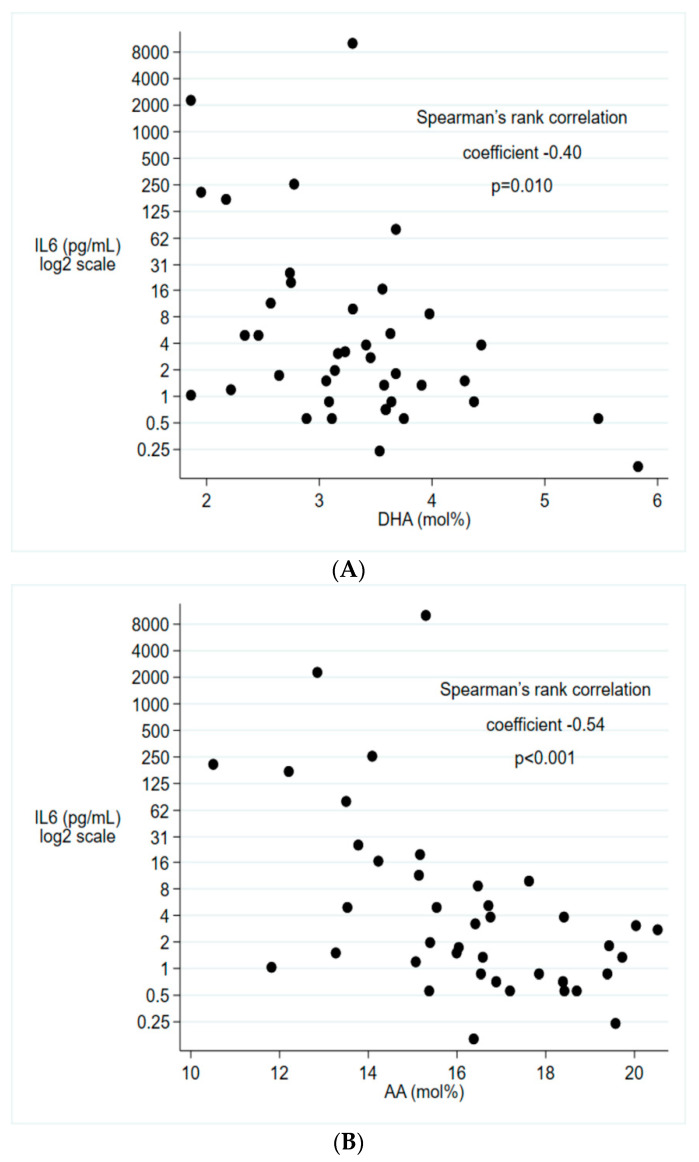
Associations between IL-6 and (**A**) DHA, (**B**) AA in cord blood.IL-6 presented on log_2_ scale. *n* = 40. Abbreviations: IL-6, interleukin-6, DHA, docosahexaenoic acid; AA, arachidonic acid.

**Table 1 nutrients-12-01996-t001:** Characteristics of the infants in the study cohort.

Characteristic	Early Systemic Inflammation *n* = 23	No Early Systemic Inflammation *n* = 67
Gestational age, weeks + days	25 + 1, 25 + 0 (23 + 1–27 + 6)	25 + 4, 25 + 3 (22 + 5–27 + 6)
Birth weight, gram	748, 760 (420–1180)	790, 760 (415–1260)
SGA	4 (17)	9 (13)
Sex, male	13 (57)	38 (57)
Preeclampsia	5 (23)	7 (11)
Cesarean delivery	9 (39)	33 (49)
Mortality before term age	5 (22)	7 (10)
Histologic Chorioamnionitis ^a^	15 (75)	33 (57)
Histologic Fetal Inflammation ^a^	10 (53)	24 (41)

Mean, median (min–max) are presented for continuous variables and number (percentage) for categorical variables. ^a^ 78 infants with available data Abbreviations: SGA, small for gestational age.

**Table 2 nutrients-12-01996-t002:** Levels of docosahexaenoic acid (DHA) and arachidonic acid (AA) in cord blood, the first and 7th postnatal day in infants with and without early systemic inflammation.

Fatty Acid	Early Systemic Inflammation, Median (25th–75th pctl)	No Early Systemic Inflammation, Median (25th–75th pctl)	Difference (95% CI) *p*-Value ^a^
DHA, cord blood ^b^ mol%	3.0 (2.2–3.3)	3.5 (2.9–3.7)	0.76 (0.1–1.5), 0.034
AA, cord blood ^b^ mol%	15.2 (12.9–16.8)	16.5 (15.1–18.4)	1.4 (−0.8–3.6), 0.216
DHA, day 1 ^c^ mol%	2.8 (2.2–3.2)	3.4 (2.8–3.9)	0.5 (0.2–0.9), 0.006
AA, day 1 ^c^ mol%	13.9 (12.0–16.1)	14.7 (13.6–15.9)	0.8 (−0.8–2.4), 0.299
DHA, day 7 ^d^ mol%	2.1 (1.9–2.4)	2.2 (1.9–2.6)	0.1 (−0.2–0.3), 0.598
AA, day 7 ^d^ mol%	7.7 (7.0–8.7)	8.0 (7.0–9.1)	0.2 (−0.7–1.1), 0.662.

^a^ Difference in median, unadjusted quantile regression; ^b^
*n* = 40 samples from cord blood; 10 infants with early systemic inflammation, and 30 infants without early systemic inflammation; ^c^
*n* = 90 blood samples at day 1; 23 infants with early systemic inflammation, and 67 infants without early systemic inflammation; ^d^
*n* = 84 blood samples at day 7; 20 infants with early systemic inflammation, and 64 infants without early systemic inflammation Abbreviations: pctl, percentile; CI, confidence interval; DHA, docosahexaenoic acid; AA, arachidonic acid.

**Table 3 nutrients-12-01996-t003:** Levels of DHA and AA in cord blood, in infants with and without histological signs of chorioamnionitis and fetal inflammation respectively.

Fatty Acid	HCA Median (25th–75th pctl)	No HCA Median (25th–75th pctl)	Difference (95% CI), *p*-Value ^a^
DHA, cord blood ^b^ mol%	3.3 (2.7–3.6)	3.6 (2.9–3.9)	0.3 (−0.3–0.9), 0.285
AA, cord blood ^b^ mol%	15.5 (14.2–17.3)	16.6 (15.4–18.4)	1.0 (−0.7–2.7), 0.230
	FIRS Median (25th–75th pctl)	No FIRS Median (25th–75th pctl)	Difference (95% CI), *p*-value ^a^
DHA, cord blood ^c^ mol%	3.2 (2.6–3.6)	3.6 (2.9–3.9)	0.4 (−0.2–1.0), 0.217
AA, cord blood ^c^ mol%	15.3 (13.8–17.6)	16.5 (15.4–18.4)	1.2 (−0.3–2.7), 0.121

^a^ Difference in median, unadjusted quantile regression; ^b^
*n* = 38 infants with information regarding HCA and cord blood; 20 infants with HCA, and 18 infants without HCA; ^c^
*n* = 37 infants with information regarding FIRS and cord blood; 15 infants with HCA, and 22 infants without FIRS Abbreviations: pctl, percentile; CI, confidence interval; HCA, histological chorioamnionitis; FIRS, fetal inflammatory response syndrome; DHA, docosahexaenoic acid; AA, arachidonic acid.

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
