# Peer review of "Docosahexaenoic Acid and Arachidonic Acid Levels Are Associated with Early Systemic Inflammation in Extremely Preterm Infants"

_nutrients, 2020, doi:10.3390/nu12071996_

Round 1

Reviewer 1 Report

This is a very interesting study that examines the association between LCPUFAs (AA and DHA) and early systemic inflammation in extremely preterm infants. The study is a secondary exploratory analysis of samples collected as part of a randomized controlled trial in which infants were randomized to receive a lipid solution with or without omega-3 LCPUFA. For this study, samples were collected from cord blood and on postnatal days 1 and 7. AA and DHA levels were compared between infants with and without early postnatal inflammation, defined as CRP ≥20 mg/L and/or IL-6 >1000 pg/mL in the first 72 postnatal hours. The results demonstrate that infants with postnatal inflammation had significantly lower DHA levels in cord blood and on the first postnatal day. The study methods and results are clearly presented, and the Discussion section is thoughtfully written. The study has some limitations, including that cord blood samples were available for only 40 of 90 infants and histopathology for 78 infants (for diagnosis of chorioamnionitis and fetal inflammation). As the authors note, it is challenging to determine the exact nature (eg, causality and timing) of the relationships between the LCPUFA levels and inflammation from these data. However, given the clinical significance of early systemic inflammation in preterm infants, the correlations between these values are interesting and advance our understanding of fatty acid exposures in the fetal and neonatal periods. I have just a few specific questions and comments for the authors:

  • How did the authors choose the definition of “early systemic inflammation?” The rationale for the CRP and IL-6 cut-offs should be described in the manuscript, as well as the reason for choosing these specific markers.
  • The authors note that the CRP levels were collected as part of routine clinical care. Was it collected in all infants within the first 3 postnatal days? If not, how many infants had CRP values and what were the clinical indications for obtaining a CRP?
  • It wasn’t entirely clear to me when the IL-6 levels were obtained- was it measured in cord blood and the first postnatal day in all infants?
  • Consider adding a supplemental table or figure to more clearly demonstrate the distribution of postnatal IL-6 and CRP values in the infants
  • It may be helpful to stratify Table 1 by infants with and without postnatal inflammation in the same manner as Table 2 to understand how clinical characteristics differed between the 2 groups

Author Response

Dear Madam/ Sir,
We are very grateful for your time and input. Please find our response to your comments in the text below, alongside details of specific revisions within the text of the manuscript.

>How did the authors choose the definition of “early systemic inflammation?” The rationale for the CRP and IL-6 cut-offs should be described in the manuscript, as well as the reason for choosing these specific markers.

Authors’ Response: The definition of “early systemic inflammation” was based on clinical samples of CRP and IL-6, routinely obtained when inserting umbilical catheter within the first two hours after birth. The cut-offs were based on clinical guidelines for suspected infection in the immediate postnatal period. We have altered the previous sentence and added one sentence aiming to clarify this on Page 2, Lines 88 – 92.

“Early systemic inflammation was defined as CRP >20 mg/L or IL-6 >1000 pg/ mL in any study sample or clinical sample within the first 72 hours from birth, with or without positive blood culture. The cut-offs were based on clinical guidelines for suspected infection in the immediate postnatal period.”

>The authors note that the CRP levels were collected as part of routine clinical care. Was it collected in all infants within the first 3 postnatal days? If not, how many infants had CRP values and what were the clinical indications for obtaining a CRP?

Authors’ Response: Clinical samples, including CRP and IL-6, were routinely obtained at insertion of central line within two hours from birth. Sampling was normally repeated within the first 72 hours after birth. In total, 85 infants had values of both CRP and IL-6 the first postnatal day and all infants had at least one value of CRP or IL-6 within the first three postnatal days. 60 infants had CRP measurements the second postnatal day, and 50 infants had CRP measurements the third postnatal day. The indication for sampling was to exclude early neonatal infection or infection induced by early instrumentation.
We have added the following paragraph on Page 2, Line 91 – Page 3, Line 95:

“During this study, IL-6 and CRP were routinely obtained in the clinic during insertion of the central line and additional sampling commonly occurred within the first 72 hours from birth. Of the 90 infants included in this study, 85 had measurements of CRP and IL-6 the first postnatal day and all infants had at least one measurement of CRP or IL-6 within the first 72 hours from birth, Supplemental Figure S1.”

>It wasn’t entirely clear to me when the IL-6 levels were obtained- was it measured in cord blood and the first postnatal day in all infants?

Authors’ Response: Thank you for pointing out that this was not clear. IL-6 was measured in cord blood according to study protocol. Measurements in cord blood were available from 40 infants. Postnatal clinical samples, used in the definition of early systemic inflammation, were obtained as previously described. All study samples of IL-6 in this paper, analyzed using multiplex assay, were obtained from cord blood only.

We have aimed to clarify this by alterations on Page 2 Line 88 – Page 3, Line 95, as previously mentioned,

on Page 3, Line 103:
“2.3. Fatty acid analyses and measurement of IL-6 in cord blood”

And on Page 3, Line 108:
“Serum concentrations of IL-6 in cord blood were measured in duplicate by …”

>Consider adding a supplemental table or figure to more clearly demonstrate the distribution of postnatal IL-6 and CRP values in the infants

Authors’ Response: Thank you for this suggestion. We have added a supplemental figure demonstrating the distribution of CRP and IL-6 the first three postnatal days.

>It may be helpful to stratify Table 1 by infants with and without postnatal inflammation in the same manner as Table 2 to understand how clinical characteristics differed between the 2 groups

Authors’ Response: Thank you. Table 1 has been stratified according to your suggestion.

Reviewer 2 Report

Hellstrom and coworkers present the results of their investigation on DHA and AA level  in preterm infants with respect to inflammatory disorders. This topic is quite important and merits great interest since it touches many areas of neonatal care. There are a number of concerns with the report:

  1. The use of SMOF or Clinoleic lipid solutions but no analysis of outcomes of each and whether they differed
  2. The analysis of phospholipids, not FFA, makes the interpretation less clear
  3. Fig 1 implies that IL6 seems to be dependent on both DHA and AA
  4. The impact of the lipid infusions on DHA and AA levels is not discussed
  5. Cord sample measurements are limited, as per authors

Further expansion on some of the points noted above provides more complete information on the study results.

Author Response

Dear Madam/ Sir,
We are very grateful for your time and input. Point-by-point responses to your comments are provided below, alongside details of specific revisions within the text of the manuscript.

>The use of SMOF or Clinoleic lipid solutions but no analysis of outcomes of each and whether they differed

Authors’ Response: In this study we aimed to analyze associations between levels of docosahexaenoic (DHA) and arachidonic acid (AA) and inflammatory conditions in the perinatal and early postnatal period. For analyses relating to levels of DHA and AA in cord blood and postnatal day 1, no parenteral nutrition had been started when these samples were collected. In table 2, fatty acid levels at postnatal day 7 were demonstrated for descriptive purposes, although this sample was collected after the defined period of systemic inflammation. At postnatal day 7 the effect of parenteral lipids on serum fatty acid levels are apparent: DHA and eicosapentaenoic acid fractions are higher in infants receiving SMOFlipid while AA is lower (Najm et al., Clin Nutr ESPEN 2017).

We agree that potential interaction between early inflammation and type of lipid emulsion in the association with later fatty acid levels would be interesting to study but it was not an objective in this paper.

>The analysis of phospholipids, not FFA, makes the interpretation less clear

Authors’ Response: It is widely accepted that fatty acids in serum/plasma phospholipid can be used as a surrogate for tissue levels, and we interpret the results based on this assumption. A recent publication also showed that the fatty acid pattern in preterm infants was uniform in several compartments (Böckmann et al., EJON 2020).

We agree that quantification of non-esterified fatty acid could have provided additional information to our analyses, but unfortunately we did not collect this lipid fraction during extraction. In a current study which includes 210 extremely preterm infants, we have collected a fraction with free fatty acids and we hope to analyze these in relation to inflammation in the future.

We have added the following sentence on Page 7, Line 254 to Page 8, Line 257:

“Quantification of non-esterified fatty acid could have provided additional information to our analyses, but it is reasonable to believe that serum phospholipid levels can be used as a surrogate reflecting tissue levels of DHA and AA in preterm infants.”

>Fig 1 implies that IL6 seems to be dependent on both DHA and AA

Authors’ Response: Yes, or at least that they are associated in cord blood. This is in line with results from our previous study (from the same cohort) showing that high AA is associated with less ROP (Löfqvist et al. JAMA Ophthalmol 2018). There are increasing evidence that DHA as well as AA are needed for the preterm infant.

We have added the following to the discussion on Page 8, Lines 258 –  261:

“As we have previously reported, infants receiving SMOFlipid showed increased serum levels of DHA but lower AA compared to infants receiving Clinoleic [16], and high levels of AA were associated with less ROP [17]. The results in this study also support the hypothesis that increasing both DHA and AA might be beneficial to the infant.”

>The impact of the lipid infusions on DHA and AA levels is not discussed

Authors’ Response: As discussed above, our main results are that DHA and AA levels are related to signs of inflammation before parenteral lipids are started. But we agree with the reviewer that also later postnatal levels in relation to type of lipid emulsion and inflammation also would be interesting to study. We have edited the paragraph in the discussion on Page 8, Lines 258 – 267, and it now reads:

“As we have previously reported, infants receiving SMOFlipid showed increased serum levels of DHA but lower AA compared to infants receiving Clinoleic [16], and high levels of AA were associated with less ROP [17]. The results in this study also support the hypothesis that increasing both DHA and AA might be beneficial to the infant. Fetal levels of DHA and AA are affected by maternal dietary intake, enzymatic activity regulating fatty acid metabolism, and placental transport [42]. After birth, infant levels are affected by the composition of omega-6 and omega-3 fatty acids in parenteral and enteral supply [43,44]. Recent recommendations state that AA should be provided to preterm infants along DHA [45]. As the exposure during both the fetal and neonatal periods are significant for infant development, it is urgent to determine optimal composition of fatty acid exposures during both these time periods.”

>Cord sample measurements are limited

Authors’ Response: We agree that this is a limitation of the study as commented on Page 7, Line 244. We hope to be able to confirm these results in a larger material.

Round 2

Reviewer 1 Report

Thank you for the opportunity to review this manuscript. The authors have addressed my comments. I have no further concerns.